# Synergistic enzymatic and bioorthogonal reactions for selective prodrug activation in living systems

Qingxin Yao[1,2], Feng Lin[3,4], Xinyuan Fan [4], Yanpu Wang[5], Ye Liu[1], Zhaofei Liu [5], Xingyu Jiang [1,2], Peng R. Chen [3,4] & Yuan Gao [1,2]

Adverse drug reactions (ADRs) restrict the maximum doses applicable in chemotherapy, which leads to failure in cancer treatment. Various approaches, including nano-drug and prodrug strategies aimed at reducing ADRs, have been developed, but these strategies have their own pitfalls. A renovated strategy for ADR reduction is urgently needed. Here, we employ an enzymatic supramolecular self-assembly process to accumulate a bioorthogonal decaging reaction trigger inside targeted cancer cells, enabling spatiotemporally controlled, synergistic prodrug activation. The bioorthogonally activated prodrug exhibits significantly enhanced potency against cancer cells compared with normal cells. This prodrug activation strategy further demonstrates high tumour inhibition efficacy with satisfactory biocompatibility, pharmacokinetics, and safety in vivo. We envision that integration of enzymatic and bioorthogonal reactions will serve as a general small-molecule-based strategy for alleviation of ADRs in chemotherapy.

[1] CAS Center for Excellence in Nanoscience, CAS Key Laboratory of Biomedical Effects of Nanomaterials and Nanosafety, National Center for Nanoscience and Technology, Beijing 100190, China. [2] University of Chinese Academy of Sciences, Beijing 100049, China. [3] Key Laboratory of Bioorganic Chemistry and Molecular Engineering of Ministry of Education, College of Chemistry and Molecular Engineering, Peking University, Beijing 100871, China. [4] Peking-Tsinghua Center for Life Sciences, Peking University, Beijing 100871, China. [5] Medical Isotopes Research Center and Department of Radiation Medicine, School of Basic Medical Sciences, Peking University, Beijing 100191, China. Correspondence and requests for materials should be addressed to P.R.C. (email: pengchen@pku.edu.cn) or to Y.G. (email: gaoy@nanoctr.cn)

Bitter pills may have wholesome effects. Anticancer drugs, for example, are highly potent but are associated with adverse drug reactions (ADRs), minimization of which is a critical but largely unmet need in cancer treatment[1,2]. The risk of side effects of these chemo-drugs on normal cell populations and specific organs (e.g. the cardiotoxicity of doxorubicin (Dox)[3]) limits the applicable dosages. This restricted dosage likely prevents the whole tumour tissue from being exposed to sufficient drug concentrations, eventually resulting in cancer recurrence and metastasis[4]. A variety of strategies have been implemented to reduce ADRs via either physical[5] or chemical control of drug activity, the latter of which is usually referred to as a prodrug, undergoing enzymatic and/or chemical transformations in situ to release the parent drug with desired pharmacological effects[6]. The development of ideal prodrugs that meet both the following requirements remains challenging: (i) targeted delivery, i.e. the delivery of effective doses of prodrugs directly to the tumour tissue, and (ii) selective activation, i.e. effective activation of prodrugs in tumour-specific environments and virtual inactivity in normal tissues. For example, antibody drug conjugates (ADCs) may often encounter non-specific activation via metabolic hydrolysis of the linker between the antibody and drug[7]. Antibody-directed enzyme prodrug therapy (ADEPT) introduces exogenous enzymes to improve the orthogonality of drug activation reactions[8]. However, the introduction of an exogenous enzyme prevents repeated administration due to issues such as immunogenicity[9]. To address the selective activation issue, the therapeutic potential of the tetrazine (Tz)–trans-cyclooctene (TCO) decaging pair has been recently exploited[10]. However, one example using this pair as an ADC linker will inevitably be limited by the insufficient penetration ascribed to mAbs[11]. Another example of the polymeric hydrogel-based application is mainly suitable for local injection to treat resectable tumours because the delivery of toxic agents is highly dependent on the method of gel administration[12].

On the other hand, recent progress in in situ enzyme-instructed supramolecular self-assembly (EISA) has demonstrated the enormous potential of this technique in targeted therapeutic applications[13–15]. Starting from small-molecular precursors that undergo enzymatic transformations to initiate the supramolecular self-assembly process, EISA is highly dependent on the activity of specific enzymes[16,17]. Because the spatiotemporal profile of up- or down-regulated enzymes is quite tumour specific[18], EISA could selectively recognize cancer cells by targeting such abnormal enzymatic activities[19]. For example, by targeting over-expressed phosphatase in HeLa cells, phosphorylated small molecules can be designed that undergo enzymatic dephosphorylation, ultimately leading to the construction of supramolecular assemblies inside live cells[20]. For cells that have very high phosphatase levels, such as Saos-2 cells, EISA may have direct and strong inhibitory effects via necroptosis[21]. By systemic administration, these small molecules may diffuse deeply into the tumour and therefore may overcome the disadvantages of insufficient penetration, which is often observed with mAbs[22].

Here, we use a combination of EISA and Tz/TCO bioorthogonal decaging reaction which simultaneously leads to spatiotemporal targeting and selective activation of prodrugs inside cancer cells, achieving the urgently needed selectivity of chemo-drugs for cancer cells over normal cells (Fig. 1). We attach a Tz moiety to the EISA motif to directly trigger the inverse-electron-demand Diels–Alder (inv-DA) reaction-mediated chemical decaging of a TCO-caged effector molecule within the intracellular environment[23–25]. Cancer cells over-expressing phosphatase exhibit intracellular EISA via the Tz-bearing NapK(Tz)YF (**3**). This intracellular EISA results in significant accumulation of Tz inside the cancer cells, enabling the liberation of a TCO-caged

prodrug (TCO-Dox) that exerts cytotoxic effects and induces cancer cell death. In contrast, in the absence of EISA, TCO-Dox is hardly activated, and normal cells are left intact. The selectivity of activated Dox for cancer cells over normal cells is enhanced 10–20-fold in comparison with that of native Dox, representing an appealing strategy for ADR reduction. Overall, our strategy enables spatiotemporally controlled, synergistic prodrug activation with superior selectivity for cancer over non-cancer cells, which ultimately leads to effective and safe tumour inhibition, as demonstrated in a xenografted cervical cancer model (HeLa cells) in mice.

## Results

**Self-assembly and prodrug activation.** The synthesis of the assembly precursor (NapK(Tz)YpF, **2**, Fig. 2a) was straightforward (see Supplementary Methods for a detailed protocol). Because KYF was determined to be an excellent self-assembly motif in a tri-peptide census survey[26], we constructed our assembling molecules based on this sequence. The conjugation of a strong π–π stacking moiety, namely, a naphthalene group, to the N terminus may further enhance the self-assembly potential of this molecule. Phosphorylation on tyrosine (Y) enabled the responsiveness of this motif to phosphatase. All of these components were sequentially conjugated via typical Fmoc-based solid-phase peptide synthesis (SPPS) to yield NapKYpF (**1**). The capability of self-assembly and hydrogelation of **1** was confirmed by the addition of 10 U mL$^{-1}$ alkaline phosphatase (ALP) into a solution of **1** (Supplementary Fig. 1). The dephosphorylated product of **1** formed transparent hydrogels with a network of uniform nanofibres (Supplementary Fig. 2), and the critical hydrogelation concentration was 1.0 mg mL$^{-1}$. The ε-amine of Lys on **1** was conjugated with the Tz derivative Tz-COOH (2-(4-(6-methyl-1,2,4,5-tetrazin-3-yl)phenyl) acetic acid) via an NHS-mediated ester–amide exchange reaction to yield **2**. Aryl-substituted Tzs are commonly used in inv-DA reactions, and a methyl group could significantly promote the decaging rate. Upon the addition of ALP (10 U mL$^{-1}$) into the solution of **2**, a transparent purple hydrogel was formed within 1 h (Fig. 2b inset), and the critical hydrogelation concentration was 1.0 mg mL$^{-1}$ (Supplementary Fig. 3). Similar to the formation of a typical hydrogel, investigation by transmission electron microscopy (TEM) confirmed the existence of numerous uniform nanofibres, each with a diameter of approximately 12 nm, which was close to that of hydrogel **1** (Fig. 2b). A further oscillatory rheological test also confirmed the formation of a stable hydrogel of **2**, as demonstrated by the following observations: (i) the dynamic storage modulus ($G'$) of the hydrogel was almost one order of magnitude higher than the dynamic loss modulus ($G''$), and (ii) the frequency dependence of the elasticity was weak (Fig. 2c and Supplementary Fig. 4).

The Tz counterpart prodrug TCO-Dox was prepared according to previously published procedures using the axial tautomer of TCO[27]. After the selection of optimized Tz and TCO, the decaging reactions in our study were further characterized (Fig. 2d). After mixing 2 μL of 10 mM TCO-Dox with 2.5 μL of 20 mM **2** in 25% acetonitrile in H$_2$O at 37 °C, the reaction mixture was analysed by ultra-performance liquid chromatography-mass spectrometry (UPLC-MS) at different time points. As shown in Fig. 2e, with **2**, TCO-Dox and Dox as references, the LC trace clearly showed the disappearance of the TCO-Dox peak 2 min after TCO-Dox was mixed with **2**. The emerging peak *a*, representing Dox, remained unchanged between 2 and 30 min, which indicated complete liberation of Dox within 2 min. The TCO adduct (peak *b*) exhibited a similar retention time as **2** (peak *c*), and these two neighbouring peaks

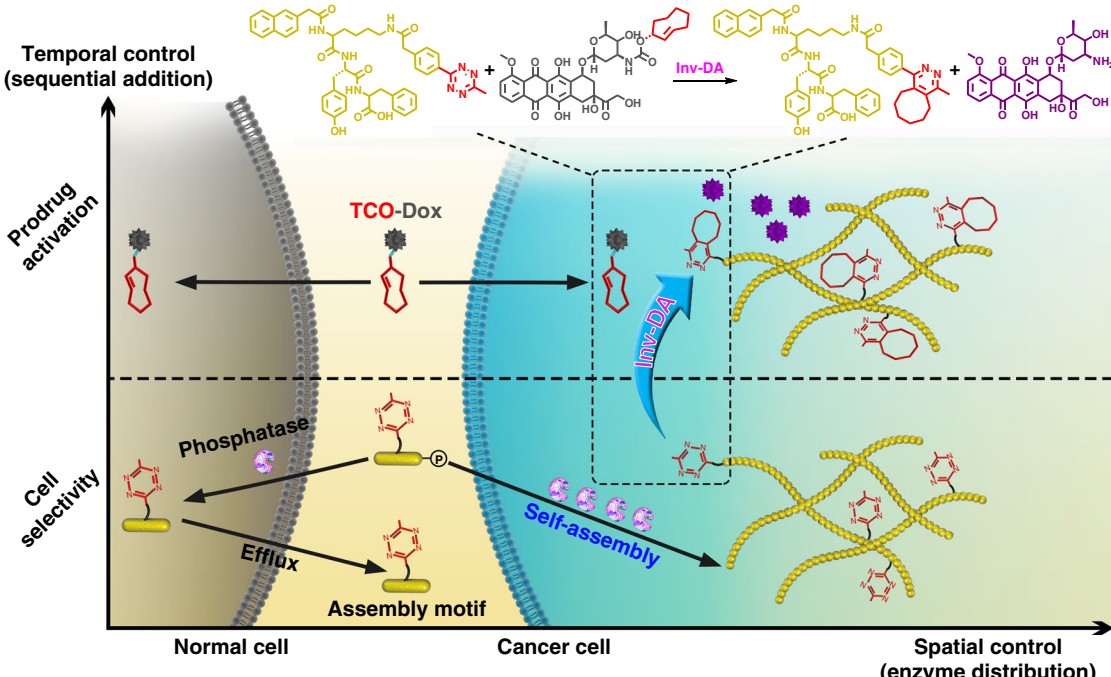

**Fig. 1** Synergistic enzymatic and biorthogonal reaction for prodrug activation in cancer cells. The process of enzyme-instructed supramolecular self-assembly selectively accumulates prodrug activation triggers in cancer cells in response to the over-expressed phosphatase. Thereafter, the prodrugs are specifically activated by the accumulated triggers through a biorthogonal reaction, which lead to cancer cells death. Inv-DA, inverse electron-demand Diels–Alder reaction

were further identified by mass spectroscopy (Supplementary Fig. 5). Similarly, we prepared TCO-caged coumarin (TCO-CMR, see Supplementary Methods for a detailed synthesis protocol)[23]. According to a previously developed fluorescence liberation[27], the fluorescence of coumarin was restored rapidly in the presence of Tz derivatives (Supplementary Fig. 6). Notably, the morphology of the nanofibres remained unchanged with the TCO adduct after liberation of Dox (Supplementary Fig. 7). Overall, we showed that the conjugation of Tz with the assembled molecules retained the decaging capability of Tz against TCO derivatives with high reactivity and liberation rates.

**Occurrence and localization of self-assembly in live cells**. Next, we tested the intracellular accumulation of Tz via EISA in an attempt to differentiate cancer cells from normal cells[28]. First, we investigated the accumulation of Tz in HeLa cells and HUVECs over time via fluorescence liberation from TCO-CMR (Fig. 3a). Each group of cells was pre-incubated with different concentrations of **2** ranging from 100 to 500 μM for 1, 2, 4, 6, 15 or 24 h. The cells were then washed with PBS thoroughly before the addition of an excessive amount of TCO-CMR (100 μL, 50 μM), and the reaction of TCO-CMR with the intracellular accumulated Tz was allowed to proceed for up to 24 h (Supplementary Fig. 8). In this experimental setting, the total liberated fluorescence was proportional to the amount of **3** accumulated inside the cells. As shown in Fig. 3b, a very small amount of fluorescence was restored in HeLa cells incubated with 100–300 μM **2**, and similar results were observed when HeLa cells were incubated with 50 or 500 μM Tz-COOH for 24 h (Supplementary Fig. 9). In contrast, the fluorescence emission increased significantly in HeLa cells pre-incubated with 400 or 500 μM **2**. Notably, when incubated with 500 μM **2**, the fluorescence in HeLa cells increased proportionately to the incubation time until reaching a maximum value at approximately 6 h. These observations suggested that (i) 500 μM **2** was sufficient for intracellular accumulation of Tz and

(ii) six hours of incubation was adequate for the accumulation to reach a saturated state. In contrast to HeLa cells, HUVECs liberated only a negligible amount of fluorescence, even in the presence of 500 μM **2**, implying that EISA barely occurred in the normal cells (Supplementary Fig. 10). Therefore, the optimized concentration of **2** (e.g. 500 μM) and incubation time (e.g. 6 h) were selected to differentiate between cancer and normal cells according to liberated fluorescence intensity (Fig. 3c). This differentiation was dependent on the considerably different phosphatase activity between these cell lines (Supplementary Fig. 11), allowing EISA to specifically occur in cancer cells as opposed to normal cells. Further incubation of HeLa cells with the phosphatase inhibitor led to a significant decrease in liberated coumarin (Supplementary Fig. 12), indicating that the accumulation of Tz in HeLa cells was positively correlated with the upregulated phosphatase activity.

In addition to the accumulation profile, we monitored the location of EISA inside cells. First, HeLa cells were incubated with 500 μM **2** for 6 h before being subjected to fractionation[29]. TEM images of the cellular fractions **M** (mitochondria, lysosomes, peroxisomes) and **P** (plasma membrane, microsomal fraction (fragments of the endoplasmic reticulum (ER)), large polyribosomes) of the treated cells showed the presence of nanofibres (Supplementary Fig. 13), verifying the occurrence of nanofibre formation inside HeLa cells. Furthermore, we used confocal fluorescence microscopy to visualize the location of EISA. HeLa cells pre-incubated with 500 μM **2** for 6 h were thoroughly washed with PBS to remove any remaining diffusive **2** or **3**. Before the addition of TCO-CMR to the cells (0 min in Fig. 3d), no blue fluorescence of coumarin was detected. Interestingly, a bright fluorescent area appeared near the nucleus within 30 s after the addition of TCO-CMR, indicating the high reactivity of the accumulated Tz anchored on intracellular assemblies (Fig. 3d and Supplementary Fig. 14). Time-course imaging showed that the fluorescence of coumarin continued to increase within the first 15 min (Supplementary Movies 1-2), indicating the abundance of Tz

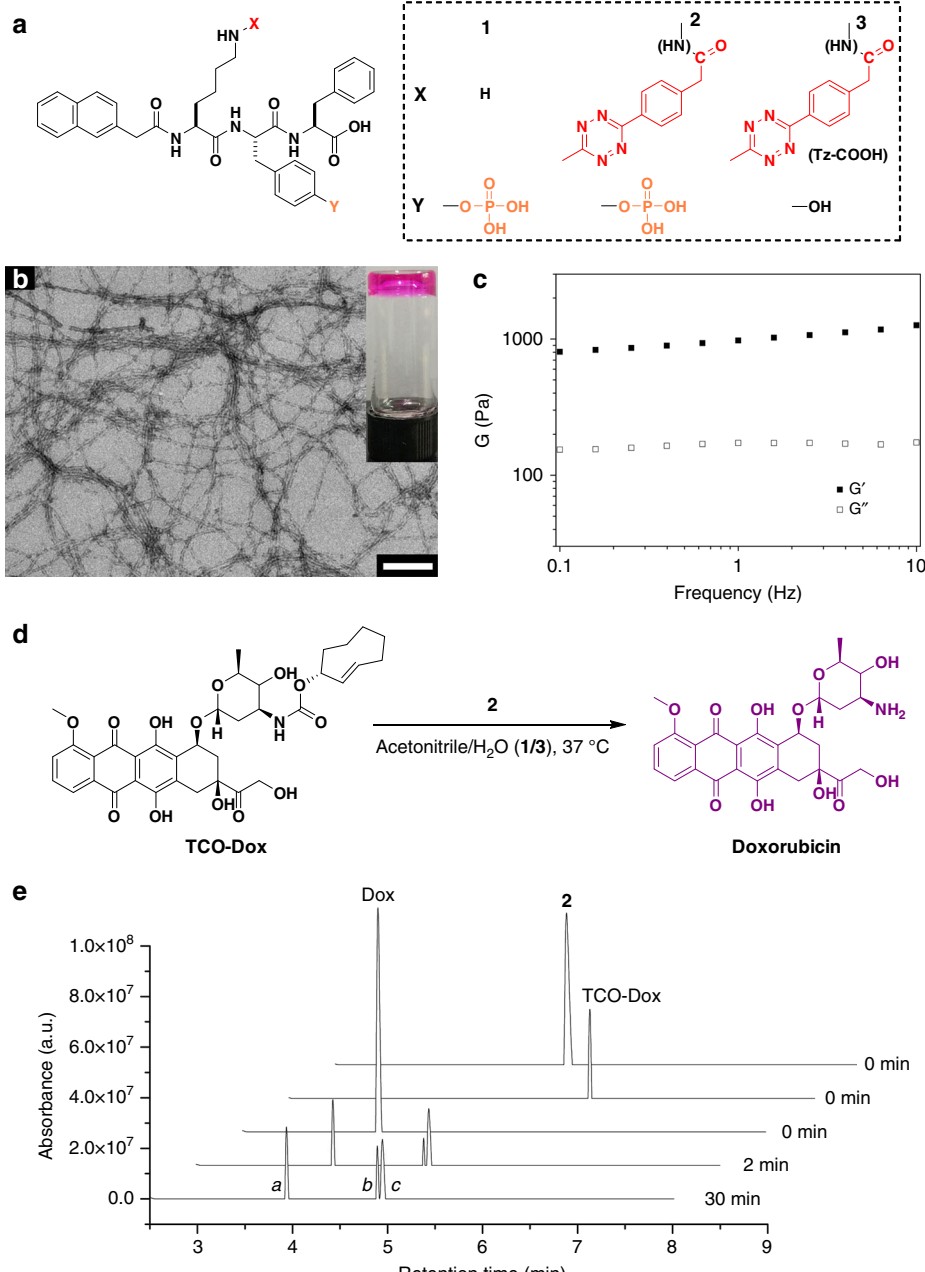

**Fig. 2** Tetrazine-bearing assembly molecules and prodrug activation in vitro. **a** Structures of NapKYpF (**1**), the tetrazine bearing self-assembly precursor NapK(Tz)YpF (**2**) and dephosphorylated assembly molecule NapK(Tz)YF (**3**). **b** TEM and optical (inset) images of hydrogel **3**. Scale bar, 200 nm. **c** Frequency sweep of the dynamic storage modulus (G′) and loss modulus (G″) of hydrogel **3**. **d** Activation of TCO-Dox by **2** in vitro. **e** UPLC trace (detected at 233 nm) recording the activation of TCO-Dox by **2** with Dox, **2**, and TCO-Dox as references (Peak *a* is Dox, peak *b* is the TCO adduct, peak *c* is **2**)

inside the cells. The blue fluorescence was near the nucleus, indicating the location of EISA inside HeLa cells (Supplementary Movie 3). Because this observation was relevant to our previous results[20], co-staining with an ER tracker (Glibenclamide BODIPY® TR, red channel) further confirmed that EISA occurred on the ER (Fig. 3d). Notably, no additional washing was required after the addition of TCO-CMR during the entire imaging process because of the low fluorescence emission from TCO-CMR. This clean background was critical for real-time visualization of fluorescence enhancement in live cells. Parallel incubation and imaging experiments were conducted with HUVECs, and only weak fluorescence was observed (Supplementary Fig. 15 and Supplementary Movie 4). This result was also consistent with previous fluorescence intensity measurements (Supplementary Fig. 10).

**Prodrug activation in live cells.** Before studying prodrug activation in EISA pre-targeted cells, we verified the biocompatibility of **2** by measuring the viability of HeLa cells and HUVECs incubated with **2**. Based on a standard MTT assay, the viability of both HeLa cells and HUVECs remained >85% after incubation with 500 μM **2** for 24 h (Supplementary Fig. 16). Based on a clonogenic assay[30], the surviving fractions of both HeLa cells and HUVECs were more than 75% after treatment with 500 μM **2** for 6 h (Supplementary Fig. 17 and Supplementary Table 1). These results proved that **2** was biocompatible and that the corresponding intracellular assemblies were eligible for further prodrug activation in live cells.

After demonstrating the biocompatibility and targeting capability of EISA, we next explored the anticancer efficacy of EISA

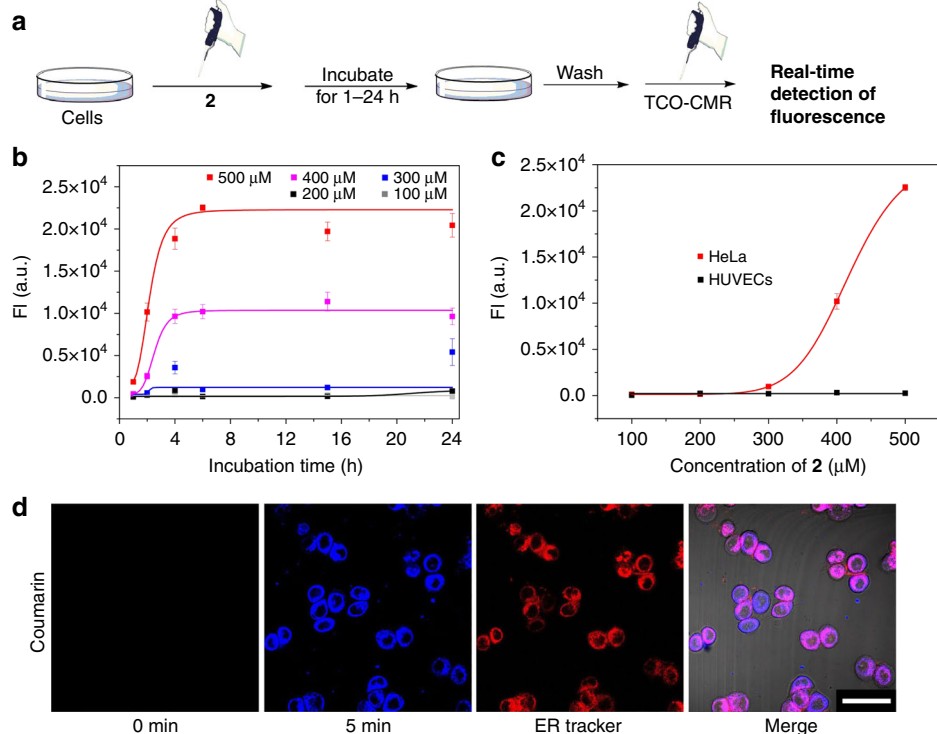

**Fig. 3** Quantification and location of the intracellular self-assembly in live cells. **a** Flowchart showing the cell culture conditions. Each group of cells was treated with different concentrations (100–500 μM) of **2** for a range of durations (1, 2, 4, 6, 15 and 24 h), washed with PBS and incubated with 100 μL of 50 μM TCO-CMR. **b** Fluorescence intensity of restored coumarin in HeLa cells for each incubation condition with 100–500 μM **2** for 1, 2, 4, 6, 15 and 24 h, respectively ($n = 6$). **c** Comparison of the fluorescence intensity from restored coumarin in HeLa cells and HUVECs. All cells were pre-treated with 100–500 μM **2** for 6 h, washed with PBS and incubated with 100 μL of 50 μM TCO-CMR for 24 h ($n = 6$). Error bars indicate standard deviation. **d** Confocal images showing the distribution of liberated coumarin (blue) and the overlap with ER tracker (red). Scale bar, 50 μm

via activation of TCO-Dox inside live cells. As a potential prodrug, we expected TCO-Dox to be less toxic than Dox until activation inside targeted cancer cells. We first investigated the cytotoxicity of Dox and TCO-Dox against HeLa cells and HUVECs. As shown in Fig. 4a, b and Table 1, TCO-Dox was much less effective at inhibiting the growth of HeLa cells than native Dox, exhibiting an $EC_{50}$ value of 6.793 μM, which was nearly 10-fold higher than that of Dox (0.732 μM). Similarly, the $EC_{50}$ of TCO-Dox against HUVECs was 5.808 μM, which was more than 40-fold higher than that of Dox (0.133 μM). The increased $EC_{50}$ value validated the reduced potency of TCO-Dox. Nonetheless, TCO-Dox and Dox exhibited no selectivity for HeLa cells and HUVECs, while the $EC_{50}$ values of both were higher against HeLa cells than against HUVECs.

The use of EISA enhanced the anticancer capability by reversing the potency of TCO-Dox against HeLa cells and HUVECs. Based on the optimized conditions obtained in the aforementioned TCO-CMR activation assay, HeLa cells were pre-incubated with 500 μM **2** for 6 h, thoroughly washed with PBS, and then incubated with TCO-Dox for 72 h. The $EC_{50}$ of activated Dox against HeLa cells was as low as 0.0298 μM, representing a 228-fold increase in cytotoxicity compared with that of the TCO-Dox prodrug alone (6.793 μM). This dramatically enhanced cytotoxicity indicated that TCO-Dox was effectively activated in HeLa cells. Moreover, the cytotoxicity of this activated Dox was approximately 25 times higher than that of native Dox ($EC_{50}$, 0.732 μM), which was likely due to the high local concentration of activated Dox in cancer cells than the administration of the native drug itself[31]. Because the **2**/**3** levels in HUVECs were negligible, the $EC_{50}$ of activated Dox was 0.272 μM against HUVECs pre-treated with 500 μM **2**, which was

approximately 9-fold higher than that against HeLa cells (0.0298 μM). To better illustrate the advantages of EISA-activated Dox, we calculated the ratio of the $EC_{50}$ value against HUVECs to that against HeLa cells (Table 2). The ratios for TCO-Dox, native Dox and EISA-activated Dox were 0.855, 0.182 and 9.128, respectively, which demonstrated the improved safety (~50-fold enhancement) of our chemically activated Dox over native Dox.

The generality of this prodrug activation strategy was further validated in additional cell lines, including Saos-2 and CCC-HEH-2 cells. Saos-2 is an osteosarcoma cell line that with extremely high expression levels of phosphatase[32], while CCC-HEH-2 is a cardiomyocyte primary cell line obtained from human embryonic myocardial tissue. Clinical data has shown severe cardiotoxicity to be induced by Dox[3,33], which is consistent with the lower $EC_{50}$ against CCC-HEH-2 cells (at 0.084 μM) than against HeLa or Saos-2 cells (Fig. 4c, d and Table 1). However, upon treatment with **2** and TCO-Dox as described previously, the $EC_{50}$ of activated Dox was 0.404 μM against CCC-HEH-2 cells, which confirmed the satisfactory tolerance. In contrast, the $EC_{50}$ of activated Dox was approximately 0.1 nM against Saos-2 cells. Notably, this low $EC_{50}$ value was contributed to not only activated Dox but also the emerging toxicity of EISA itself[21]. As the MTT assay with Saos-2 cells showed relatively low starting levels (Fig. 4c), we further tested the cytotoxicity of **2** against Saos-2 cells by both the MTT assay and clonogenic assay (Supplementary Figs. 16 and 17 and Supplementary Table 1). The results indicated that while the assemblies were compatible with the other cell lines that we tested, they were toxic to Saos-2 cells to some extent, which was likely due to the extremely high phosphatase activity in this cell type (Supplementary Fig. 11).

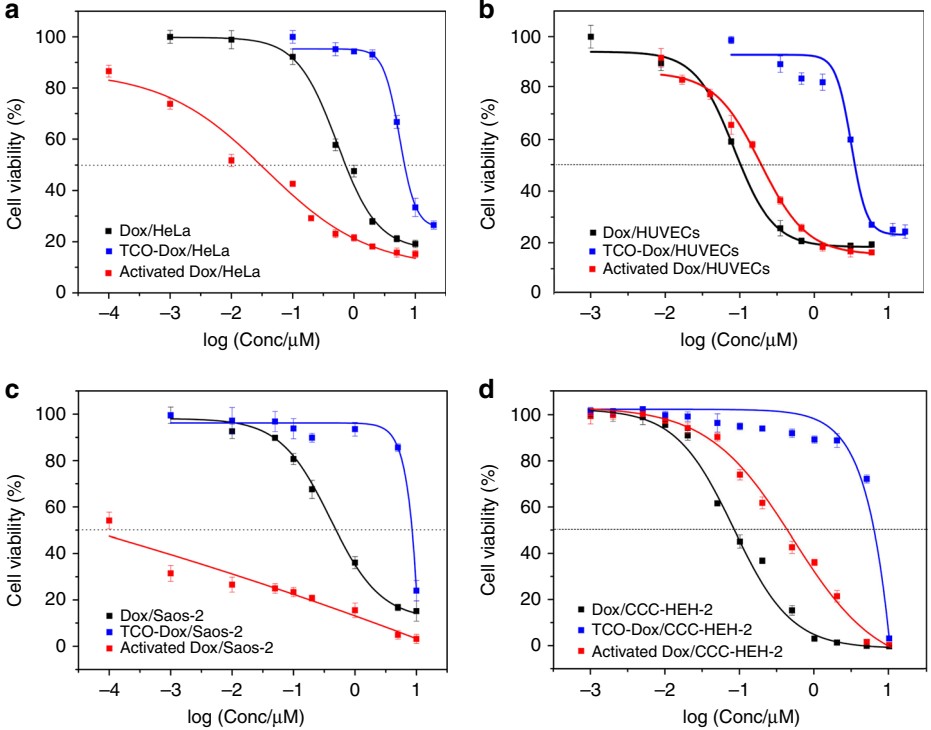

**Fig. 4** Prodrug activation with high selectivity towards cancer cells. Comparison of the 72 h cytotoxicity of TCO-Dox, Dox and activated Dox against **a** HeLa cells, **b** HUVECs, **c** Saos-2 cells and **d** CCC-HEH-2 cells. Cytotoxicity profiles of Dox and TCO-Dox were obtained by incubating cells with different concentrations of each compound for 72 h. For the cytotoxicity profile of activated Dox, cells were pre-incubated with 100 μL of 500 μM **2** for 6 h followed by incubation with 200 μL of TCO-Dox at different concentrations for 72 h ($n = 6$). Error bars indicate standard deviation

**Table 1 Calculated EC$_{50}$ values for Dox, TCO-Dox and activated Dox against each cell line**

| Drug | EC$_{50, \text{HeLa}}$ | EC$_{50, \text{Saos-2}}$ | EC$_{50, \text{CCC-HEH-2}}$ | EC$_{50, \text{HUVECs}}$[a] |
|---|---|---|---|---|
| Dox | 0.732 | 0.425 | 0.084 | 0.133 |
| TCO-Dox | 6.793 | 7.569 | 6.169 | 5.808 |
| Activated Dox[b] | 0.029 | 0.0001 | 0.404 | 0.272 |

[a]EC$_{50}$ values are determined the 72 h cytotoxicity of each drug. The units of EC$_{50}$ are in μM.
[b]Activated Dox indicates that each cell line was pre-incubated with 100 μL of 500 μM **2** for 6 h and incubated with 200 μL of TCO-Dox at different concentrations for 72 h.

**Table 2 Calculated EC$_{50}$ ratios of Dox, TCO-Dox and activated Dox against each cell line**

| Drug | CCC-HEH-2/HeLa | CCC-HEH-2/Saos-2 | HUVECs/HeLa | HUVECs/Saos-2 |
|---|---|---|---|---|
| Dox | 0.115 | 0.198 | 0.182 | 0.313 |
| TCO-Dox | 0.908 | 0.815 | 0.855 | 0.767 |
| Activated Dox | 13.557 | 4040 | 9.128 | 2720 |

Overall, EISA-activated Dox exhibited high toxicity against cancer cells but not against normal cells (Table 2).

In addition, we demonstrated that EISA formation was essential for the enhanced cytotoxicity of TCO-Dox observed in cancer cells. Similar to the above drug efficacy test, HeLa cells and HUVECs were pre-incubated with **2** at a concentration of 300 μM for 6 h (EISA was hardly observed inside cells under these conditions, Fig. 3c) before the addition of TCO-Dox for

measurement of EC$_{50}$ values. As shown in Supplementary Fig. 18, the EC$_{50}$ against HeLa cells was 1.199 μM, which was approximately 40-fold higher than that of group incubated with 500 μM **2** (EC$_{50}$ = 0.0298 μM). Consistent with the coumarin liberation results, the sharp increase in the EC$_{50}$ value was most likely due to the relatively low accumulation of Tz in HeLa cells. For HUVECs, the EC$_{50}$ value against TCO-Dox varied little (0.367 μM to 0.272 μM) when the concentration of **2** increased from 300 μM to 500 μM, indicating the wide selectivity window of our EISA for cancer cells over normal cells.

**In vivo anticancer efficacy**. To evaluate the tumour specificity and in vivo distribution of **2**, a mixture of 50 mg kg$^{-1}$ **2** and 0.875 mg kg$^{-1}$ $^{125}$I-labelled **2** was intravenously (i.v.) injected into BALB/c nude mice bearing subcutaneous HeLa tumours, which were followed by single-photon emission computed tomography/computed tomography (SPECT/CT) imaging. Figure 5a shows the distribution of **2/3** at different time points. Within the first hour, **2/3** was mainly accumulated in the tumour and liver. Then, **2/3** was gradually cleared and almost removed from the mouse body within 24 h, guaranteeing the safety of this assembly. In the control group injected with 0.875 mg kg$^{-1}$ $^{125}$I-labelled **2** alone, almost no **2/3** was accumulated in the tumour site (Fig. 5b), indicating that EISA did not occur in the tumour in the absence of sufficient assembly precursors.

Next, we applied UPLC-MS/MS to examine the pharmacokinetics[34] of i.v. administered **2** and the corresponding dephosphorylated **3** in blood, liver and tumour samples from tumour-bearing mice (Fig. 5c). Due to the presence of phosphatases in blood, **2** was partially transformed to the dephosphorylated product **3** in plasma (Supplementary Fig. 19). However, in contrast to the rapid clearance of **2/3** from blood (nearly complete 2 h post injection), the total concentrations of **2/3** in tumours

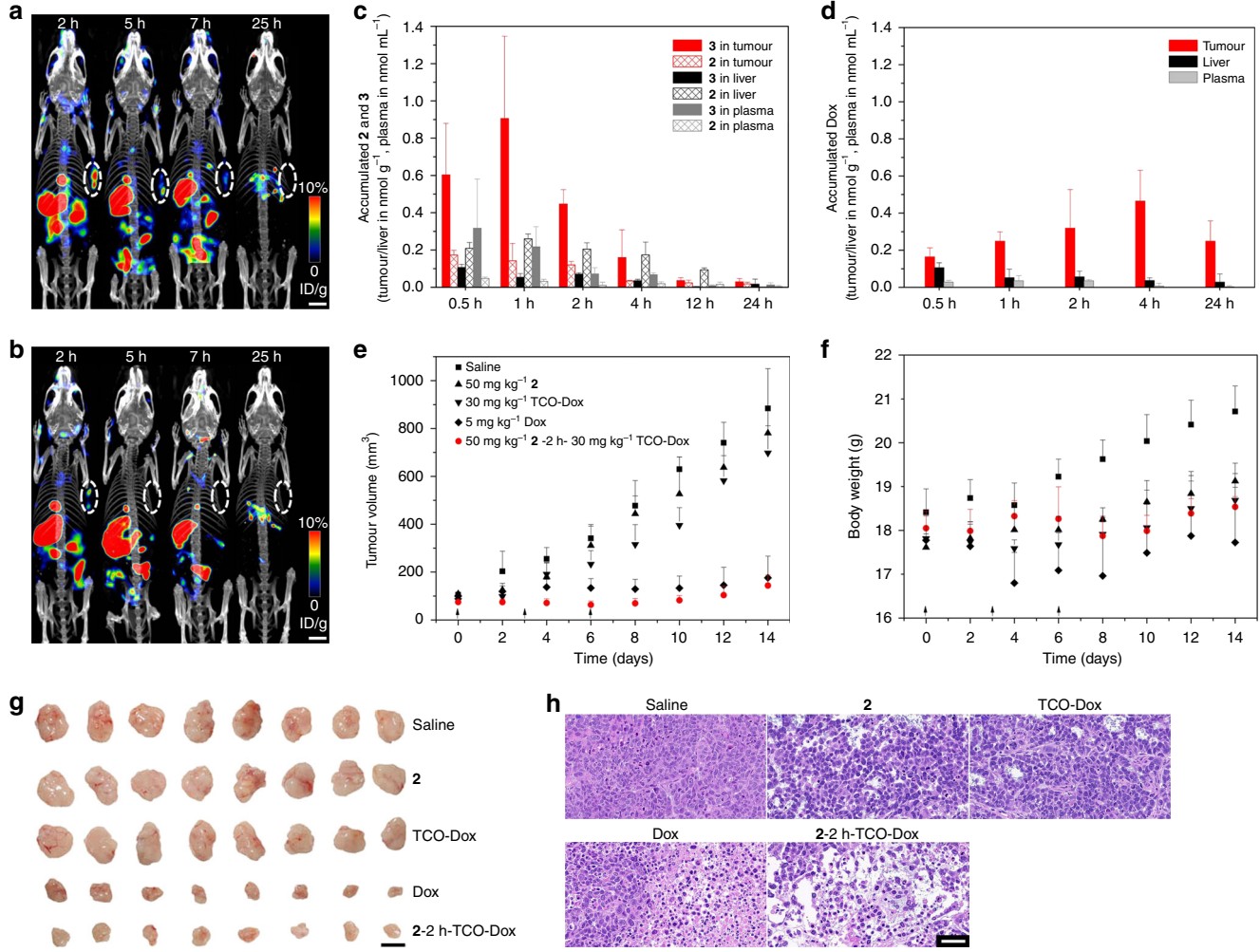

**Fig. 5** Biodistribution of self-assemblies and in vivo prodrug activation to inhibit tumour growth. **a**, **b** SPECT/CT imaging demonstrated the distribution of **2** at different time points after i.v. administration of **a** a mixture of 50 mg kg$^{-1}$ **2** and 0.875 mg kg$^{-1}$ $^{125}$I-labelled **2** (700 μCi) and **b** 0.875 mg kg$^{-1}$ $^{125}$I-labelled **2** (700 μCi) alone. Tumour sites are highlighted by white circles. Scale bars, 5 mm. **c** UPLC-MS/MS analysis of the pharmacokinetics of **2** and **3** in the tumour, liver and plasma after i.v. injection of 50 mg kg$^{-1}$ **2** (n = 5). The results demonstrated the selective accumulation of **3** in tumours over liver or plasma. **d** UPLC-MS/MS analysis of activated Dox in tumour, liver and plasma after treatment [i.v. injection of 50 mg kg$^{-1}$ **2** followed by i.v. injection of 30 mg kg$^{-1}$ TCO-Dox after a 2-h interval] (n = 5). The results demonstrated the effective enrichment of activated Dox in tumours over liver or plasma. **e** Tumour growth curves for mice after i.v. injection with saline, 50 mg kg$^{-1}$ **2**, 30 mg kg$^{-1}$ TCO-Dox, 5 mg kg$^{-1}$ Dox and 50 mg kg$^{-1}$ **2**-2 h-30 mg kg$^{-1}$ TCO-Dox (n = 8). The arrows on the x-axis indicate treatment duration in days. **f** Quantification of mouse weights in each group of (**e**). Error bars indicate standard deviation. **g** Photograph of dissected tumour samples after treatment for 14 days. Scale bar, 10 mm. **h** Representative histological examination results of the dissected tumours using HE staining. Scale bar, 50 μm

were maintained at high levels for the first 2 h. In particular, the concentration of **3** in tumours increased significantly within 1 h, indicating specific accumulation in tumours over liver or blood, which is attributed to the phosphatase upregulation of HeLa cells. To avoid unwanted activation of the TCO-Dox prodrug in blood, we chose a 2-h interval between the administration of **2** and TCO-Dox. As shown in Fig. 5d, TCO-Dox was effectively activated in tumours to provide a high and sustained concentration of Dox (at least 24 h after injection of TCO-Dox), which was dependent on tumour-specific accumulation of **2/3** and relatively high concentration of TCO-Dox remaining in the blood (Supplementary Fig. 20). In comparison, the concentration of activated Dox remained very low in the liver, and it was almost undetectable in the blood within 24 h (Fig. 5d). For example, the concentration of activated Dox in tumours was >10-fold higher than that in liver or blood 4 h after TCO-Dox administration, indicating the high tumour specificity of our EISA-enabled prodrug activation strategy.

Finally, to investigate the antitumour efficacy in vivo, HeLa tumour-bearing BALB/c nude mice were randomly allocated into five groups (n = 8) when the tumour volume reached 80–100 mm³. The treatment group received a dose of 50 mg kg$^{-1}$ **2** and a dose of 30 mg kg$^{-1}$ TCO-Dox after 2-h intervals (abbreviated as **2**-2 h-TCO-Dox). Variations in the dose of **2** or TCO-Dox or the interval may slightly curtail the overall tumour inhibition efficacy (Supplementary Fig. 21). For comparison, the remaining four control groups received 50 mg kg$^{-1}$ **2**, 30 mg kg$^{-1}$ TCO-Dox, 5 mg kg$^{-1}$ Dox or saline. After three courses of treatment, tumour growth was markedly inhibited in the **2**-2 h-TCO-Dox treatment group (Fig. 5e). The tumour volume in the saline control group increased rapidly. The tumours treated with 50 mg kg$^{-1}$ **2** or 30 mg kg$^{-1}$ TCO-Dox alone exhibited rapid growth similar to that observed in the saline control group. For the group that received native Dox, although tumour growth could be inhibited with 5 mg kg$^{-1}$ Dox, the body weight decreased substantially (Fig. 5f). Further increase in the Dox dose resulted in mouse death

(Supplementary Fig. 22), indicating the systemic toxicity of Dox in mice. In contrast, the body weights of mice in the **2**-2 h-TCO-Dox group remained steady after treatment, indicating the satisfactory biosafety of our strategy.

After treatment for 14 days, the mice were sacrificed, and tumours were dissected (Fig. 5g) for histological examination with haematoxylin–eosin (HE) staining. As shown in Fig. 5h, the **2**-2 h-TCO-Dox treatment group exhibited necrotic lesions of the tumours, exhibiting cell shrinkage, loss of contact and coagulation, which was similar to the tumours treated with Dox. These observations indicated that tumour cells can be efficiently destroyed by activated Dox. However, histopathological images of the tumours treated with saline, **2** or TCO-Dox demonstrated that the HeLa cells remained normal. To further assess possible systemic toxicity, the major organs were harvested for histological analysis. The **2**-2 h-TCO-Dox treatment group did not exhibit any observable differences compared to the saline control group based on histological examination of the heart, liver, spleen, lung and kidney (Supplementary Fig. 23), further indicating the good biocompatibility and safety of our prodrug strategy.

## Discussion

We have developed a small-molecule-based strategy for anticancer drug delivery with the advantage of ADR alleviation. The concept of integrating enzymatic reaction-triggered supramolecular self-assembly and bioorthogonal cleavage reaction-triggered chemical decaging enabled spatiotemporally controlled prodrug activation. Specifically, by targeting cancer cells over-expressing phosphatase, including HeLa and Saos-2 cells, Tz-bearing assembly precursors were efficiently dephosphorylated and self-assembled due to increased hydrophobicity. The formation of these assemblies significantly increased the local concentration of the assembly molecules with high levels of accumulation of Tz moieties. This bioorthogonally reactive handle was able to react with TCO and release the caged cargo. As shown in this study, intracellular self-assembly led to high levels of accumulation of Tz in cancer cells, which guaranteed rapid and specific activation of TCO-Dox and facilitated target engagement of the activated Dox inside the nucleus. The overall results showed not only a great increase in the potency of Dox against malignant cells but also the satisfactory selectivity for cancer over non-cancer cells, which indicated the synergistic effects of these two approaches in the alleviation of ADRs. Animal studies further validated not only the biosafety of our EISA compounds and TCO-Dox prodrug but also the excellent in vivo tumour inhibition within a clinically relevant setting. Notably, the dose applied in the current study was below the maximum tolerant dose (MTD) of Dox. Therefore, maximal therapeutic effect may be achieved with a higher dose of TCO-Dox by increasing the number of injections. Given that the TCO-caging method is compatible with various functional groups[35], our strategy presented here could be applied to other cytotoxic drugs or toxins that are currently unsuitable for medical use, which may significantly expand the scope of anticancer therapeutic agents. Finally, the subcellular location of the EISA-triggered accumulation of the assemblies (e.g. near microtubules) could be leveraged to bring certain toxins (e.g. maytansine) in close proximity to the corresponding cellular targets with further enhancement of therapeutic effects.

## Methods

**EISA in live cells**. All cells were purchased from the National Infrastructure of Cell Line Resource (Beijing, China). HeLa cells, HUVECs and CCC-HEH-2 cells were cultured in a humidified $CO_2$ (5%) incubator at 37 °C in DMEM supplemented with 10% foetal bovine serum (FBS, Gibco) and 1% Pen Strep (PS, Gibco). Saos-2 cells were cultured in a humidified $CO_2$ (5%) incubator at 37 °C in McCoy's 5A medium supplemented with 10% FBS and 1% PS. Cells were plated in flat-bottom 96-well plates (Corning) at a density of $10^4$ cells per well and allowed to attach for

12 h. Then, the culture medium was replaced with 100 μL of pre-warmed medium containing **2** at different concentrations (100–500 μM). After incubation for different durations (1–24 h), the cells were rinsed with PBS, and then 100 μL of 50 μM TCO-CMR (dissolved in PBS) was added for real-time detection of the fluorescence intensity of coumarin in a plate reader (Perkin Elmer).

**Live cell imaging**. Cells were first placed in a glass chamber and cultured with 2 mL of culture medium containing 500 μM **2** for 6 h. Then, the cells were rinsed with PBS and incubated with pre-warmed staining solution (500 nM ER-Tracker™ Red dye) for 30 min at 37 °C. The cells were rinsed again with PBS at least five times. After the cell-containing glass chamber was fixed on the confocal microscope stage, 1 mL of 50 μM TCO-CMR (dissolved in PBS) was added, and fluorescent images/movies were captured immediately.

**Cell viability**. Cells were plated in flat-bottom 96-well plates at a density of $3 \times 10^3$ cells per well. After 12 h of cell attachment, the cell culture medium was replaced with 100 μL of pre-warmed media containing the compounds at different concentrations. After 24–72 h of incubation, cell proliferation was assessed by the MTT assay. A fresh mixture of 10 μL of thiazolyl blue tetrazolium bromide (MTT) and 90 μL of medium were freshly added to each well. After incubation for 4 h, the medium was gently removed. The formazan crystals that formed were dissolved in 110 μL of DMSO, and the absorbance was subsequently measured with a plate reader (Perkin Elmer) at 490 nm. $EC_{50}$ values were calculated using Origin software. To measure the cytotoxicity of **2**, HeLa cells, HUVECs, Saos-2 cells and CCC-HEH-2 cells were incubated with 100 μL of 1–500 μM **2** for 24 h, and then, cell viability was determined. To measure the cytotoxicity of TCO-Dox or Dox, HeLa cells, HUVECs, Saos-2 cells or CCC-HEH-2 cells were incubated with 200 μL of 0.001–10 μM TCO-Dox or Dox for 72 h, and the cell viability was determined. To determine the cytotoxicity of activated Dox, HeLa cells, HUVECs, Saos-2 cells or CCC-HEH-2 cells were pre-incubated with 100 μL of 500 μM **2** for 6 h, and then, **2** was removed and 200 μL of 0.0001–10 μM TCO-Dox was added for 72 h of incubation, and cell viability was determined.

**Tumour models**. All animal studies were performed in accordance with the Institutional Animal Use and Care Regulations approved by the National Institutes of Health Clinical Center/Animal Care and Use Committee (NIH CC/ACUC). Athymic female nude mice (5–6 weeks old) were purchased from Beijing Huafukang Bioscience Co., Ltd., and raised in a specific-pathogen-free barrier facility. Tumour-bearing mice were established by subcutaneously injecting a suspension of $10^6$ HeLa cells per mouse into the front flank of each nude mouse.

**SPECT/CT imaging**. The tumour-bearing mice were used for SPECT/CT imaging once the tumour volume reached approximately 200 mm³. For the preparation of $^{125}I$-labelled **2**, 60 μg of **2** was dissolved in 150 μL of phosphate buffer (pH = 7.0) in a vial coated with 20 μg of iodogen (Sigma, St. Louis, MO). Then, 9 μL of Na$^{125}I$ (2.4 mCi) (Beijing Atom High Tech, Beijing, China) was added, and the mixture was kept at room temperature for 1 h until the labelling ratio reached more than 95%. For SPECT/CT imaging, a mixture of 50 mg kg$^{-1}$ **2** and 0.875 mg kg$^{-1}$ $^{125}I$-labelled **2** (700 μCi) was i.v. administered to investigate the distribution of **2/3**, and 0.875 mg kg$^{-1}$ $^{125}I$-labelled **2** (700 μCi) was administered as a control. After anaesthetization with 2% isoflurane in oxygen, SPECT and helical CT scans of the mice were performed at 2, 5, 7 and 25 h on a NanoScan SPECT/CT imaging system (Mediso, Budapest, Hungary).

**Pharmacokinetic analyses**. When the tumour volume reached approximately 200 mm³, five mice in each group were used for pharmacokinetic analyses of compound **2/3** and for in vivo prodrug activation. For the pharmacokinetic study of compound **2/3**, mice received an i.v. injection of 50 mg kg$^{-1}$ **2**. For in vivo prodrug activation, mice received an i.v. injection of 50 mg kg$^{-1}$ **2** and a subsequent 30 mg kg$^{-1}$ TCO-Dox after a 2-h interval. After different time intervals, blood samples were collected from the tail veins, and then, the tumours and livers were harvested for UPLC-MS/MS analysis.

**Tumour inhibition test**. When the tumour volume reached approximately 80–100 mm³, the mice were divided into five groups ($n = 8$) and used for therapeutic studies. Mice in the treatment group received an i.v. injection of 50 mg kg$^{-1}$ **2** and a subsequent 30 mg kg$^{-1}$ TCO-Dox after a 2-h interval. In the control groups, mice were received 50 mg kg$^{-1}$ **2**, 30 mg kg$^{-1}$ TCO-Dox, 5 mg kg$^{-1}$ Dox or saline. The mice received three treatment courses on day 0, day 3 and day 6. Tumour size and body weight were measured every 2 days. After 14 days, the mice were sacrificed. Tumours and main organs were obtained for histological examination with HE staining.

**Reporting summary**. Further information on research design is available in the Nature Research Reporting Summary linked to this article.

## Data availability

A reporting summary for this article is available as a Supplementary Information file. Data that support the findings reported herein are available on reasonable request from the corresponding author.

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

## Acknowledgements

This work was supported by National Key R&D Program of China (2017YFA0205901 to Y.G. and 2016YFA0501501 to P.R.C.), NSFC (21675036 to Y.G. and 21521003 to P.R.C.) and Hundred Talents Project of the Chinese Academy of Sciences for financial support. We also thank F.F. He (National Center for Nanoscience and Technology) for developing the UPLC-MS/MS analysis methods.

## Author contributions

Q.Y., Y.G. and P.R.C. developed the concepts and designed the experiments; Q.Y., F.L., X. F. and Y.W. performed the experiments; Q.Y., Y.G. and P.R.C. wrote the paper; Q.Y., F. L., Y.L., Z.L., X.J., Y.G. and P.R.C. analysed results and edited the manuscript.

## Additional information

**Competing interests:** The authors declare no competing interests.

