## [Peer Review File · Nature Communications]

Reviewers' comments:

Reviewer #1:

The manuscript of Yuan Gao and col. describes an interesting concept that combines bioorthogonal and enzymatic reactions in order to "synergistically" deliver drugs into tumor. In principle this idea could justify a publication in Nature Communication, I find that the experimental data do not fully support the claims made by authors.

As minor point, the introduction is to my opinion out of focus. For instance, cardiotoxicity is mentioned as a main limitation of doxorubicin, while no cardiotoxicity model was run by authors, also references to the ADC's lack of tumor penetration seems not relevant to this paper since the authors have not demonstrated that their approach could in a way or another overcome this issue... Generally speaking in this manuscript authors tend often to go beyond the demonstration they have made. This should really be carefully checked.

The discussion about spatiotemporal profile of up- and down- regulated enzyme, in cancer, that is key for this paper, is too superficial. In particular the differential expression phosphatase in HeLa cell vs other cell types is not precisely demonstrated and discussed. Then author should also discriminated enzyme expression and enzymatic activity. This brings me to the first important point, the lack of demonstration that accumulation of **2** in the cells (and the tumors) is linked to phosphatase activity. I have seen no control with phosphatase inhibitor, siRNA or turn-on probe reflecting real phosphatase activity. This is even more important since there is no proof that the differences observed between the different cell lines is due to EISA effect and not on simple difference in cell permeation, cell excretion or cell division half time

A clear table with enzyme expression/ enzymatic activity vs EISA effect should be provided (if possible also on tumor homogenate).

Generally speaking I also regret the scarce experimental descriptions that make the manuscript difficult to review. Conditions used for in solution study of hydrogel formation and those used in cell culture are difficult to correlate in term of concentration and solvent exposition time. A precise experimental part for all the experiment described in the different figures should be provided in the SI. For example, no experimental protocol on how fiber from S7 was generated can be find in the SI.

Minor point, decaging was done in ACN/H₂O why did the author not select buffer instead of water? Fiber S7 was done in DMSO, what is the effect of DMSO on fibers?

Concerning cell viability: (Fig 4 and S16), activated cells seem to suffer, how does author explain the lower starting level of MTT assays? (description of MTT cell assays are a bit imprecise concerning the incubation time of **2** and Dox-TCO, the densities of HeLa and HUVEC Saos). I would really suggest to add one experimental description per figure. In addition, I would strongly suggest a clonogenic assay to check the effect of **2** on cell. I would also suggest to including the imaging in cell experiment of labeled fiber with precise experimental conditions.

For this part also there are many imprecise claims as for instance... page 7, "its corresponding intracellular assembly" ... no proof of intracellular assembly is provided, "after demonstrating... the targeting capability of EISA" targeting is not demonstrated.... The concept of "selectivity" (P.8) is not clearly defined (I am not sure that it is "therapeutically" relevant) and ref 30 is missing.

Also P.8 the discussion about liberation of Doxorubicine close the Nucleus seems to me a bit fussy and do not really help understanding this paper. This point could be discussed

in a separated paper with precise localization of labeled fiber (which in the for reviewer only material) do not clearly support above mentioned assertion

Concerning the in vivo experiments: as for other parts claims are often imprecise "2 was gradually metabolized" no proof of metabolization is provided. Most annoyingly biodistribution images show that compound 2 remain in the organism at 7h. While TCO-Dox is injected after 2 h. Authors did not rule out the fact that reaction proceeds in plasma. Such in vivo drug inactivation *via* click chemistry has been described recently (Wagner and col. Nat. Comm. 2017, 8, 15242). I would suggest performing at least some PK studies in order to show that no activation proceeds in the circulation, liver or other organs where 2 seems to accumulate. (This unspecific accumulation also calls back to my remark on the misleading claim of "targeting capability of EISA" see above). Similarly metabolic (or at least plasmatic and microsomal) stability of compound 2 and TCO-Dox have not been reported. There is thus no clear evidence that drug activation proceeds indeed within the tumor as claimed by authors. This seems even more important since a "slow release process" could also explain the tumor growth inhibition effect especially if one consider the described two stage PK profile of Dox.

In addition, if I am right, 50mg/kg of 2 IV roughly correspond to 1 mg/mL max plasmatic dose. (This is close from 1mg/mL critical hydrogelation concentration). Exposition of tumor seems thus quite different from what was done in cell culture experiment. Authors should discuss this point in relation with recommended PK studies. Without such studies and differential organ/tumor excretion of 2 there seems to be no evidence of the preferential accumulation of 2 in tumor.

In conclusion I think that this paper describes an interesting strategy that could be published in Nat. Comm. However, claims should be restricted to what is demonstrated, experimental part should be more precise and comprehensive, cell lines enzymatic activity should be studied, accordingly in vitro control experiments should be added, clonogenic experiment should be done at least for compound 2, precise PK studies should be included and activation reaction should be studied in vivo using LC/MS.

Reviewer #2 (Remarks to the Author):

The manuscript of Chen, Gao and coworkers describes the combined use of enzyme-instructed self-assembly (EISA) and of a trusted bioorthogonal reaction for in vivo imaging and therapy. While these concepts have already been described, the authors show that it is the combination of both that enables a novel prodrug based therapeutic regimen. The proposed temporal decoupling of loading cells with a prodrug-converting phosphorylated tetrazine (Tz) molecule (2) (that in turn is converted into EISA hydrogel-forming compound 3) followed by administration of a less toxic Doxorubicin (Dox) prodrug (TCO-Dox) allowed boosting both the efficiency of Dox-based cytotoxicity and selectivity for the phosphatase-overexpressing tumor cells. This concept of spatiotemporally-controlled prodrug activation is appealing as it may result in better cancer therapeutic options, while diminishing adverse drug reaction.

In a first step the authors demonstrate the formation of Tz-bearing nanofibers by EISA and the efficiency of TCO-Dox activation by Tz-containing assembly molecule 2-mediated decaging in vitro. Using a dye-linked TCO (TCO-CMR) the best concentration applied on cells for an efficient bioorthogonal decaging is determined. Further, the selectivity of three tumor cell lines over normal HUVEC cells is measured in vitro for Dox alone, TCO-Dox and the activated Dox. Clearly, Dox activation works with good selectivity for all three tumor cell lines. It would have been nice to include a further normal cell line for this analysis, yet it may not be necessary since EISA is already well established.

Following in vitro experiments the concept is tested in vivo in HeLa-bearing Balb/c mice. Accumulation of iodinated Tz-assembly molecules in the tumor is evidenced by SPECT/CT-monitoring indicating the formation of EISA nanofibers in the HeLa tumor (effectuated only at a high administered dose, providing further evidence for the EISA-based deposition).

The authors then perform a therapy experiment in mice with Dox alone (5 mg/kg) and prodrug TCO-Dox (30 mg/kg, activated and non-activated) with 3 injections at day 0,3,6, leading to a significant tumor growth inhibition for Dox and activated TCO-Dox. While the MTD for Dox is clearly shown to be 5 mg/kg, the MTD for TCO-Dox is not given, which may allow using even higher doses of TCO-Dox, hence possibly leading to complete tumor eradication.

Remarks:

- p 3, last sentence of Introduction: please specify more clearly "...on a mouse model"
- p 4, l 6, "arrayed" sounds a bit strange to me
- p 4, second line from below: "By mixing..." : The Volume (i.e. the conc.) should be given (or at least in the materials and methods)
- p5, l 20 "...and an excessive amount of TCO-CMR..." better give molar excess
- p6, l 1, "(likely in the form of nanofibers)" Should be removed (or proven).
- p6, l 9 "...dependent on the drastically varied phosphatase activity...in cancer cells..." Please be more specific. It would also be nice to introduce the varied phosphatase activity and the application to specific cancer cell lines in the Introduction.
- p6, 7th line from below: Is Ref. 26 here OK?
- p9, l 7, "Although a trace amount of EISA may be trapped in..." sounds strange to me
- p9, l 13, "...injected a mixture of..." what mixture, a 1:1? And how much was administered?

Figures:

The Figures are nice, yet some fonts are too small.

Fig 2e: Absorbance was measured at what wavelength? a,b,c should be described in the caption.

Fig 3a,b: The legends (insets?!) should be better readable. b) At which timepoint was the accumulation determined?

Fig 5: caption: give the tumor model, describe what is circled in the SPECT, the administration arrows are hardly visible. For better comparison, FigureS17 should be integrated into Fig 5.

Figure S8,S9: This was measured with TCO-CMR? Please write in the caption.

Discussion:

- The effect of the residence time of the nanofibers in the cells (how long is the high local

concentration maintained?) should be discussed.

- Is a similar cell-uptake of TCO-CMR and TCO-Dox to be expected?
- It could be discussed if the maximal therapeutic effect was already reached (or if higher conc. of TCO-Dox, more injections, combination of the method with other EISA-based approaches that target different cellular components might be useful).

In summary, the presented experiments are well-controlled and the authors convincingly demonstrate the potential of their novel strategy for therapeutic purpose. I can therefore recommend the manuscript for publication in Nat Commun provided that the points mentioned above are met.

Our point-by-point responses to each specific question from our reviewers are shown below (for clarity, the original comments are shown in *italic*).

Reviewer: #1

[Comments: The manuscript of Yuan Gao and col. describes an interesting concept that combines bioorthogonal and enzymatic reactions in order to "synergistically" deliver drugs into tumor. If in principle this idea could justify a publication in Nature Communication, I find that the experimental data do not fully support the claims made by authors.]

Response: Thanks for the comments. We have followed this reviewer's comments and performed additional experiments to further support our claim. Please see the following responses for more details.

[1) As minor point, the introduction is to my opinion out of focus. For instance, cardiotoxicity is mentioned as a main limitation of doxorubicin, while no cardiotoxicity model was run by authors, also references to the ADC's lack of tumor penetration seems not relevant to this paper since the authors have not demonstrated that their approach could in a way or another overcome this issue... Generally speaking in this manuscript authors tend often to go beyond the demonstration they have made. This should really be carefully checked.]

Response: We have carefully checked the manuscript and revised the description to match what we have done.

In addition, since the clinical studies have confirmed that doxorubicin induced cardiotoxicity is mediated by topoisomerase-II β in cardiomyocytes (*Nat. Med.*, **2012**, *18*, 1639), we have tested the compatibility of our prodrug activation strategy with the cardiomyocyte CCC-HEH-2 cells. As shown in Figure 4d, the results confirmed that our prodrug activation procedure was safer than the direct usage of native doxorubicin (e.g. a 4.8-fold increase of EC₅₀) on cardiomyocyte cells.

For the tumor penetration issue, since we did not focus on this question in our manuscript, we have omitted the statement "the issues of insufficient penetration or internalization due to the intrinsic size of mAb" in the revised manuscript.

[2) The discussion about spatiotemporal profile of up- and down- regulated enzyme, in cancer, that is key for this paper, is too superficial. In particular the differential

expression phosphatase in HeLa cell vs other cell types is not precisely demonstrated and discussed. Then author should also discriminated enzyme expression and enzymatic activity. This brings me to the first important point, the lack of demonstration that accumulation of 2 in the cells (and the tumors) is linked to phosphatase activity. I have seen no control with phosphatase inhibitor, siRNA or turn-on probe reflecting real phosphatase activity. This is even more important since there is no proof that the differences observed between the different cell lines is due to EISA effect and not on simple difference in cell permeation, cell excretion or cell division half time

A clear table with enzyme expression/ enzymatic activity vs EISA effect should be provided (if possible also on tumor homogenate).]

Response: We have determined the phosphatase activity in different cell lines including HeLa, Saos-2, HUVECs and CCC-HEH-2 cells. The absolute phosphatase activities in different cell lines were shown in the figure below, with the two cancerous cell lines (HeLa and Saos-2) possessing a much higher phosphatase activity than the normal cells (HUVECs and CCC-HEH-2). This result has now been included as Figure S11 in Supporting Information.

Figure S11. The phosphatase activity of different cell lines used in this study. For each of the four cell lines (HeLa, Saos-2, HUVECs and CCC-HEH-2 cells), 3×10^5 cells were plated on a 6-well plate for 6 h before 150 μ L cell lysis buffer (1.0 mol/L PMSF in RIPA, Beijing Solarbio Science and Technology Co., Ltd) was added. Cell samples were then centrifuged at 10000 g for 5 min and the supernatant was collected for the detection of phosphatase activity *via* a standard protocol using para-nitrophenyl phosphate (pNPP) as chromogenic substrate and para-nitrophenol as the reference.

Further, to verify the relationship between the intracellular accumulation of **2** (actually should be the tetrazine-bearing assembling molecules) and the phosphatase activity, we investigated the effects of the phosphatase inhibitor on the coumarin release experiment in both HeLa and HUVECs cells. As shown below, the addition of phosphatase inhibitor to HeLa cells showed a 91.1% decrease of coumarin release than without inhibitor treatment. In contrast, the coumarin release in HUVECs remained unchanged with and without the phosphatase inhibitor treatment. This result indicated that the accumulation of tetrazine-bearing assembling molecules was well correlated with the phosphatase activity in the cell. This result has now been included as Figure S12 in Supporting Information.

Figure S12. The effect of phosphatase inhibitor on the accumulation of tetrazine-bearing assembling molecules. The addition of phosphatase inhibitor significantly decreased the accumulation of assembling molecules in HeLa cells. Cells were plated on flat bottom 96-well plates at a density of 10^4 cells/well and allowed to attach for 6 h before being incubated with 100 μ L fresh culture medium w/o 25 μ M phosphatase inhibitor cocktail (Beyotime Biothchnology) for 12 h. Cells were next treated with 100 μ L 500 μ M of **2** (w/o 25 μ M phosphatase inhibitor cocktail) for 1, 2, 4, 6, 15, 24 h, respectively. The culture medium was then removed, rinsed by PBS buffer, and incubated with 100 μ L 50 μ M of TCO-CMR was added for another 24 h. Finally, the fluorescence intensity of released coumarin was measured on a plate reader.

[3) Generally speaking I also regret the scarce experimental descriptions that make the manuscript difficult to review. Conditions used for in solution study of hydrogel

formation and those used in cell culture are difficult to correlate in term of concentration and solvent exposition time. A precise experimental part for all the experiment described in the different figures should be provided in the SI. For example, no experimental protocol on how fiber from S7 was generated can be find in the SI.]

Response: We have added the detailed descriptions for all experiments as enclosed in **Methods, Supplementary Methods** and the captions for each figure.

[4) Minor point, decaging was done in ACN/H₂O why did the author not select buffer instead of water? Fiber S7 was done in DMSO, what is the effect of DMSO on fibers?]

Response: In the decaging experiment, we initially chose ACN/H₂O mixed solvent to match the HPLC mobile phase. To determine if the choice of buffer may make a difference, we performed the decaging of TCO-Dox in PBS buffer instead of water. The results below showed a similar decaging efficiency in water and in buffer conditions:

UPLC trace recorded activation of TCO-Dox by 2 with doxorubicin, 2, and TCO-Dox as the references (the three traces marked with “0 min”). After mixing TCO-Dox by **2** for 2 min, the trace marked with “2 min” gave three peaks (*a*) doxorubicin, (*b*) TCO adduct and (*c*) **2**. This result was almost identical to that of Figure 2e, making the choice of water or buffer equivalent in decaging. (Detailed protocol: 2.5 μ L 20 mM of **2** was mixed with 96.5 μ L acetonitrile and 300 μ L PBS. After the solution was equilibrated at 37 $^{\circ}$ C, 2 μ L 10 mM of TCO-Dox was added and the solution was thoroughly mixed and incubated at 37 $^{\circ}$ C in the dark.)

In terms of TCO-CMR decaging, we chose the DMSO/H₂O mixed solvent according to the published standard protocol (*Angew. Chem. Int. Ed.* **2016**, *55*, 14046-14050). Figure S7 discussed a minor point that the tetrazine bearing nanofibers remained after reaction with TCO-CMR. This experiment was done in PBS buffer without the addition of DMSO. The experimental detail has been added in the figure caption.

[5) Concerning cell viability: (Fig 4 and S16), activated cells seem to suffer, how does author explain the lower starting level of MTT assays? (description of MTT cell assays are a bit imprecise concerning the incubation time of 2 and Dox-TCO, the densities of Hela and HUVEC Saos). I would really suggest to add one experimental description per figure. In addition, I would strongly suggest a clonogenic assay to check the effect of 2 on cell.]

Response: A detailed experimental description has been included. As for the lower starting level of MTT assays in Figure 4 and S16, we further extended the concentration of TCO-Dox to 10⁻⁴ μM and the updated Figure 4a, 4c and Figure S18 were shown below.

Figure 4. Comparison of the 72 h cytotoxicity of TCO-Dox, Dox and activated Dox against (a) HeLa and (c) Saos-2 cells.

Figure S18. The cytotoxicity of TCO-Dox against HeLa cells and HUVECs pre-treated with different concentrations of 2.

The viability of HeLa cells started around 90% which was a normal starting level in MTT assay. In contrast, the starting level of MTT assay on Saos-2 cells remained below 60%, which implied that the assemblies were toxic to Saos-2 cells to some extent. To further investigate this, we performed a 24 h cytotoxicity study of 500 μM of **2** against Saos-2 cells (The result was enclosed in Figure S16). The result was consistent with the observation in drug activation experiment in Figure 4c. The emerging cytotoxicity of the assemblies against Saos-2 can be attributed to the outstanding phosphatase activity of Saos-2 cells (reported by the reference *J. Am. Chem. Soc.* **2016**, *138*, 3813-3823, and also measured by us in Figure S11). This result has now been included as Figure S16 in Supporting Information.

Figure S16. The 24 h cytotoxicity study of **2 against different cell lines used in this study.** Each of the four cell lines (HeLa, HUVECs, Saos-2 and CCC-HEH-s cells) at a density of 3×10^3 cells/well were incubated with 100 μL pre-warmed medium containing **2** at different concentrations for 24 h. Cell proliferation was then assessed by MTT assay.

In addition, the clonogenic assay has been performed according to the reviewer's suggestion. Consistent with our MTT assay, the clonogenic assay also showed that pre-treatment of 500 μM **2** for 6 h had certain cytotoxicity on Saos-2 cells but not on HeLa, HUVECs or CCC-HEH-2 cells. This result has now been included as Table S1 in Supporting Information and we have added further discussion regarding this observation in the manuscript.

Table S1: Clonogenic assay on HeLa, HUVECs, Saos-2 and CCC-HEH-s cells pre-treated with 500 μ M of 2 for 6 h.

Cells	Plating Efficiency	Surviving Fraction
HeLa	42.5 \pm 3.1%	75.3 \pm 1.9%
Saos-2	25.5 \pm 2.6%	0 \pm 0%
HUVECs	47.4 \pm 3.2%	96.2 \pm 3.6%
CCC-HEH-2	61.8 \pm 3.8%	92.2 \pm 8.3%

[6] *I would also suggest to including the imaging in cell experiment of labeled fiber with precise experimental conditions.]*

Response: We have performed additional cell fractionation experiments to address this comment. Please see point (7).

[7] *For this part also there are many imprecise claims as for instance... page 7, "its corresponding intracellular assembly" ... no proof of intracellular assembly is provided,]*

Response: Thanks for the reviewer's comment. The intracellular assembly can be proved from the TEM images of cell fractions (as shown below) and the STORM images which are now included as For Review Only Materials.

Figure S13. TEM images of the cellular fractions of HeLa cells pre-incubated with 2. Cells were incubated with 500 μ M of 2 for 6 h before being subject to cell lysis and fractionation. The TEM images of the fractions **M** and **P** showed the

existence of nanofibers, suggesting the formation of the nanofibers inside HeLa cells. Detailed protocol: HeLa cells were fractionated and divided into four parts, **N**, nuclei; pellet sample **M**, mitochondria, lysosomes, peroxisomes; pellet sample **P**, plasma membrane, microsomal fraction (fragments of ER), large polyribosomes; pellet sample **R** and supernatant sample **C** (R+C), ribosomal subunits, small polyribosomes, soluble portion of cytoplasm (Cytosol).

[8) "after demonstrating... the targeting capability of EISA" targeting is not demonstrated....]

Response: Thanks for pointing out this. We originally intended to conclude the previous cell experiments which showed the selective formation of EISA within cancer cells. To make this statement clearer, we revised this sentence:

Original:

After demonstrating the biocompatibility and targeting capability of EISA...

Revised:

After demonstrating the biocompatibility of **2** against normal cells and the selective formation of EISA within cancer cells...

[9) The concept of "selectivity" (P.8) is not clearly defined (I am not sure that it is "therapeutically" relevant) and ref 30 is missing.]

Response: We originally intended to compare the safety between native drug and activated drug and defined the “selectivity” which turned out to be less therapeutically relevant. We have now removed the definition of “selectivity” and directly compared their EC₅₀ values.

The reference numbering has also been corrected.

[10) Also P.8 the discussion about liberation of Doxorubicine close the Nucleus seems to me a bit fussy and do not really help understanding this paper. This point could be discussed in a separated paper with precise localization of labeled fiber (which in the for reviewer only material) do not clearly support above mentioned assertion]

Response: We agree with the reviewer’s comments. We have deleted the description of “liberation of Doxorubicin close the Nucleus”. Further precise localization study of labeled fiber will be reported and discussed in a separated paper.

[II) Concerning the *in vivo* experiments: as for other parts claims are often imprecise "2 was gradually metabolized" no proof of metabolization is provided. Most annoyingly biodistribution images show that compound 2 remain in the organism at 7h. While TCO-Dox is injected after 2 h. Authors did not rule out the fact that reaction proceeds in plasma. Such *in vivo* drug inactivation via click chemistry has been described recently (Wagner and col. *Nat. Comm.* 2017, 8, 15242). I would suggest performing at least some PK studies in order to show that no activation proceeds in the circulation, liver or other organs where 2 seems to accumulate. (This unspecific accumulation also calls back to my remark on the misleading claim of "targeting capability of EISA" see above). Similarly metabolic (or at least plasmatic and microsomal) stability of compound 2 and TCO-Dox have not been reported. There is thus no clear evidence that drug activation proceeds indeed within the tumor as claimed by authors. This seems even more important since a "slow release process" could also explain the tumor growth inhibition effect especially if one consider the described two stage PK profile of Dox.]

Response: Thanks for the comments. We have performed i) serum stability study of compound 2 and TCO-Dox, ii) pharmacokinetics study of 2, the dephosphorylated 3, TCO-Dox as well as *in vivo* activated Dox by using UPLC-MS/MS. The results confirmed that the tumor indeed gained much higher concentration of activated Dox comparing to liver or plasma. Whereas the concentration of Dox within tumor was significantly increased in the first 4 h, the concentration of Dox in liver and plasma decreased constantly. These results supported that i) prodrug activation mainly occurred inside tumor rather than other organs or plasma; ii) the "slow release" mechanism in which Dox was activated before entering tumor sites can be excluded. Detailed results are shown below:

(1) Pharmacokinetic study of compound **2**, **3** and *in vivo* activated Dox.

Figure 5. (c) LC-MS/MS analysis of the pharmacokinetics of **2** and **3** in tumor, liver and plasma after i.v. injection of **2** (50 mg/Kg, n=5). The results clearly demonstrated the selective accumulation of **3** in tumor than in liver or plasma. (d) LC-MS/MS analysis of activated Dox in tumor, liver and plasma after the treatment [*i.v. injection of 2* (50 mg/Kg), 2 hours interval, then *i.v. injection TCO-Dox* (30 mg/Kg)] (n=5). The results clearly demonstrated the enrichment of activated Dox in tumors than in liver or plasma.

(2) Stability of compound **2** and TCO-Dox in human serum

Figure S19. Serum stability study of compound (a) **2** and (b) TCO-Dox. 10 µM of compound **2** or TCO-Dox were dissolved in human serum (n=5). At each time point, the remaining **2**, corresponding generated **3** and remaining TCO-Dox were determined by UPLC-MS/MS. The decrease of **2** roughly equals to the increase of **3** in the first 6 hours due to dephosphorylation. Both compounds then underwent degradation in serum afterwards.

(3) Pharmacokinetics of TCO-Dox.

Figure S20. Plasma concentration of TCO-Dox. Intravenous injection of 30 mg/Kg of TCO-Dox was conducted on mice (n=5) and the blood samples were collected from tail vein at selected time intervals with the remaining TCO-Dox determined by UPLC-MS/MS.

[12] In addition, if I am right, 50mg/kg of **2** IV roughly correspond to 1 mg/mL max plasmatic dose. (This is close from 1mg/mL critical hydrogelation concentration). Exposition of tumor seems thus quite different from what was done in cell culture experiment. Authors should discuss this point in relation with recommended PK studies. Without such studies and differential organ/tumor excretion of **2** there seems to be no evidence of the preferential accumulation of **2** in tumor.]

Response: Based on the rough estimation (e.g. the nude mice body weight of 20.0 g and blood volume of 2.0 mL), the 50 mg/kg dosage corresponded to 0.5 mg/mL. The critical hydrogelation concentration is based on **3** not on **2**. Therefore, the injected **2** will not self-assemble until *in situ* dephosphorylation as a result of high phosphatase activity such as in HeLa tumor cells.

(1) PK study showed an increased accumulation of **2** and **3** in tumor than in liver and blood (especially for **3**), indicating the preferred accumulation of tetrazine bearing molecules in tumor.

(2) Practically, we have tried lower dosage of **2** (such as 20 mg/Kg of **2**) to test the therapy efficacy. However, the tumor inhibition effect was not as good as 50mg/kg. This result has been included as Figure S21 in Supporting Information.

Reviewer: #2

The authors then perform a therapy experiment in mice with Dox alone (5 mg/kg) and prodrug TCO-Dox (30 mg/kg, activated and non-activated) with 3 injections at day 0,3,6, leading to a significant tumor growth inhibition for Dox and activated TCO-Dox. While the MTD for Dox is clearly shown to be 5 mg/kg, the MTD for TCO-Dox is not given, which may allow using even higher doses of TCO-Dox, hence possibly leading to complete tumor eradication.]

Response: We appreciate the suggestion of MTD which may yield a complete tumor eradication. We will tackle the MTD issue in another paper against a more challenging tumor model in a future work.

[1) Remarks:- p 3, last sentence of Introduction: please specify more clearly "...on a mouse model"]

Response: We have specified this description as “on a xenografted cervical cancer model (HeLa cells) in mice.”

[- p 4, l 6, "arrayed" sounds a bit strange to me]

Response: We changed “arrayed” into “sequentially conjugated”

[- p 4, second line from below: "By mixing..." : The Volume (i.e. the conc.) should be given (or at least in the materials and methods)]

Response: The volume and concentration of each compound are now included in the manuscript and SI.

[- p5, l 20 "...and an excessive amount of TCO-CMR..." better give molar excess]

Response: The exact volume and concentration of TCO-CMR are now given in the manuscript.

[- p6, l 1, "(likely in the form of nanofibers)" Should be removed (or proven).]

Response: We have removed this description.

[- p6, l 9 "...dependent on the drastically varied phosphatase activity...in cancer cells..." Please be more specific. It would also be nice to introduce the varied

phosphatase activity and the application to specific cancer cell lines in the Introduction.]

Response: To make this statement more specific, we have supplemented the phosphatase activity of each cell lines used in this study in Figure S11. In addition, we added one sentence to introduce the varied phosphatase activity and the application to Saos-2 cells, which is correlated to our following results in the manuscript.

[- p6, 7th lane from below: Is Ref. 26 here OK?]

Response: We have carefully checked the reference list and correct the citation mistakes.

[- p9, l 7, "Although a trace amount of EISA may be trapped in..." sounds strange to me]

Response: This description has been removed.

[- p9, l 13, "...injected a mixture of..." what mixture, a 1:1? And how much was administered?]

Response: The mixture contained 50 mg/Kg of **2** and 0.875 mg/Kg of ¹²⁵I labelled **2**. We have supplemented the experimental details in **Methods**.

[2) Figures: The Figures are nice, yet some fonts are too small.

Fig 2e: Absorbance was measured at what wavelength? a,b,c should be described in the caption.]

Response: We have revised the figures accordingly. The traces were recorded at 233 nm. The description of *a,b,c* has been added in the figure caption.

[Fig 3a,b: The legends (insets?!) should be better readable. b) At which time point was the accumulation determined?]

Response: We have revised the legends in Figure 3a, b. We determined the accumulation when the cells were incubated with **2** for 6 h. The detailed information has been added in the figure caption.

[Fig 5: caption: give the tumor model, describe what is circled in the SPECT, the administration arrows are hardly visible. For better comparison, FigureS17 should be

integrated into Fig 5.]

Response: We have added the tumor model, described the circles, and added the control SPECT/CT images as Fig. 5b.

[Figure S8,S9: This was measured with TCO-CMR? Please write in the caption.]

Response: We have revised the captions for Figure S8-9 accordingly.

[3) Discussion:

-The effect of the residence time of the nanofibers in the cells (how long is the high local concentration maintained?) should be discussed.]

Response: According to the CMR release experiments in which HeLa cells were treated with **2** for varied incubation periods (Fig. 3b), the high local concentration of assembling molecules can be maintained from 6 h to 24 h.

[- Is a similar cell-uptake of TCO-CMR and TCO-Dox to be expected?]

Response: This is an interesting point. Neutral coumarin such as TCO-CMR is cell permeable. As of doxorubicin, though the exact internalization pathways remain debated, it is believed that doxorubicin enter cells *via* passive diffusion (*Pharmac. Ther.* **1982**, *18*, 293-311; *Cancer Lett.* **1985**, *28*, 213–221). Thus, it is reasonable that the neutral doxorubicin such as TCO-Dox may also simply diffuse inside cells. Overall, we interpret that both TCO-CMR and TCO-Dox diffuse inside cells and the uptake should be linear to the concentration applied. To further validate this assumption, we will make an explicit measurement in the future studies.

[- It could be discussed if the maximal therapeutic effect was already reached (or if higher conc. of TCO-Dox, more injections, combination of the method with other EISA-based approaches that target different cellular components might be useful).]

Response: We have discussed these aspects in the Discussion session.

REVIEWERS' COMMENTS:

Reviewer #1 (Remarks to the Author):

The manuscript of Professors Goa and Chen has improved a lot, they have addressed most of my concern. In particular they have added many precisions about experimental condition, formation detailed discussion about pharmaco-kinetic issues, supporting experiments that demonstrate that supramolecular self-assembly is enzymatically instructed, they have restricted their claims to what was demonstrated in the paper.

Some questions about critical concentration for self-assembly keep puzzling me, but it is beyond the scope of this paper and will be addressed in more biophysical oriented paper.

To my opinion the paper is now fully suitable for publication in Nature communication.

Reviewer #2 (Remarks to the Author):

Looking at the point-to-point reply and the added material I was pleased to see that the manuscript has been thoroughly revised by the authors, essentially, all key points that were raised during the first reviewing have been appropriately addressed.

In particular, the authors provide new experimental data that support their statements:

- the crucial role of phosphatase activity for the claimed EISA-effect was further investigated by measuring activity in two cancer and two normal cell lines, and by looking at the tetrazine compound accumulation in cells in the presence/absence of phosphatase inhibitor.
- additional experiments on cell viability have been performed on the four cell lines in the presence of caged/natural Dox, and different (high) conc. of phosphorylated tetrazole compound 2. A clonogenic assay is now provided.
- The intracellular location of the EISA nanofibers has been revealed and discussed.
- The serum stability of 2 and TCO-Dox have been measured.
- Importantly, a pharmacokinetic study and a basic biodistribution study have been performed, which support the authors' claims (included now in Fig 5)

Some minor points:

- p3, line 4. "Notably, these...." This sentence may be re-written, e.g. "By systemic administration these small molecules may diffuse deeply into the tumor and therefore may overcome the disadvantages of insufficient penetration, which is often observed with mAbs."
- Phosphatase upregulation of cancer cells may be discussed further regarding the therapeutic potential of this novel prodrug technology.

The requested changes in the text have been made and the points raised have been discussed satisfactorily. The manuscript now also reads clearer, since all necessary experimental details are now given.

As the proposed concept of spatiotemporally-controlled prodrug activation is novel and appealing for cancer therapeutic options, I support its publication in Nat Commun.

Our point-by-point responses to each specific question from our reviewers are shown below (for clarity, the original comments are shown in *italic*).

Reviewer #1 (Remarks to the Author):

[The manuscript of Professors Gao and Chen has improved a lot, they have addressed most of my concern. In particular they have added many precisions about experimental condition, formation detailed discussion about pharmaco-kinetic issues, supporting experiments that demonstrate that supramolecular self-assembly is enzymatically instructed, they have restricted they claims to what was demonstrated in the paper.

Some questions about critic concentration for self-assembly keep puzzling me, but it is beyond the scope of this paper and will be addressed in more biophysical oriented paper.

To my opinion the paper is now fully suitable for publication in Nature communication.]

Response: Thanks for the reviewer's positive comments.

Reviewer #2 (Remarks to the Author):

[Looking at the point-to-point reply and the added material I was pleased to see that the manuscript has been thoroughly revised by the authors, essentially, all key points that were raised during the first reviewing have been appropriately addressed.

In particular, the authors provide new experimental data that support their statements:

- the crucial role of phosphatase activity for the claimed EISA-effect was further investigated by measuring activity in two cancer and two normal cell lines, and by looking at the tetrazine compound accumulation in cells in the presence/absence of phosphatase inhibitor.

- additional experiments on cell viability have been performed on the four cell lines in the presence of caged/natural Dox, and different (high) conc. of phosphorylated tetrazole compound 2. A clonogenic assay is now provided.

- The intracellular location of the EISA nanofibers has been revealed and discussed.

- The serum stability of 2 and TCO-Dox have been measured.

- Importantly, a pharmacokinetic study and a basic biodistribution study have been

performed, which support the authors' claims (included now in Fig 5)]

Response: Thanks for the reviewer's positive comments.

[Some minor points:

- p3, line 4. "Notably, these...." This sentence may be re-written, e.g. "By systemic administration these small molecules may diffuse deeply into the tumor and therefore may overcome the disadvantages of insufficient penetration, which is often observed with mAbs."]

Response: Thanks for the reviewer's advice. We have re-written the sentence.

Original:

Notably, these small molecules can diffuse deeply inside tumor via a systemic administration that can potentially overcome the disadvantages of insufficient penetration of mAb

Revised:

By systemic administration these small molecules may diffuse deeply into the tumor and therefore may overcome the disadvantages of insufficient penetration, which is often observed with mAbs.

[- Phosphatase upregulation of cancer cells may be discussed further regarding the therapeutic potential of this novel prodrug technology.]

Response: Thanks for the reviewer's advice. We have discussed the "phosphatase upregulation of cancer cells" in the biodistribution of **2/3** section.

[The requested changes in the text have been made and the points raised have been discussed satisfactorily. The manuscript now also reads clearer, since all necessary experimental details are now given.

As the proposed concept of spatiotemporally-controlled prodrug activation is novel and appealing for cancer therapeutic options, I support its publication in Nat Commun.]

Response: Thanks for the reviewer's positive comments.